# Natural Product Aloesin Significantly Inhibits Spore Germination and Appressorium Formation in *Magnaporthe oryzae*

**DOI:** 10.3390/microorganisms11102395

**Published:** 2023-09-26

**Authors:** Guohui Zhang, Rongyu Li, Xiaomao Wu, Ming Li

**Affiliations:** 1Institute of Crop Protection, College of Agriculture, Guizhou University, Guiyang 550025, China; 13985809124@163.com (G.Z.); wuxm827@126.com (X.W.); 2College of Life and Health Science, Kaili University, Kaili 556000, China

**Keywords:** aloesin, *Magnaporthe oryzae*, conidia, appressorium formation, enzyme activity, proteomics

## Abstract

This study aims to determine the effects of the natural product aloesin against *Magnaporthe oryzae*. The results exposed that aloesin had a high inhibitory effect on appressorium formation (the EC_50_ value was 175.26 μg/mL). Microscopic examination revealed that 92.30 ± 4.26% of *M. oryzae* spores could be broken down by 625.00 μg/mL of aloesin, and the formation rate of appressoria was 4.74 ± 1.00% after 12 h. *M. oryzae* mycelial growth was weaker than that on the control. The enzyme activity analysis results indicated that aloesin inhibited the activities of polyketolase (PKS), laccase (LAC), and chain-shortening catalytic enzyme (Aayg1), which are key enzymes in melanin synthesis. The inhibition rate by aloesin of PKS, LAC, and Aayg1 activity was 32.51%, 33.04%, and 43.38%, respectively. The proteomic analysis showed that actin expression was downregulated at 175.62 μg/mL of aloesin, which could reduce actin bundle formation and prevent the polar growth of hyphae in *M. oryzae*. This is the first report showing that aloesin effectively inhibits conidia morphology and appressorium formation in *M. oryzae*.

## 1. Introduction

*Magnaporthe oryzae* remains the major pathogen of rice crops, affecting rice production negatively across many different geographical regions [1,2,3,4]. Disease management approaches have primarily focused on synthetic chemicals and disease-resistant rice varieties [5,6]. However, these synthetic chemicals are expensive, difficult to obtain for small-scale farmers, and have detrimental effects on the environment, farmers’ and consumers’ health, and beneficial predators and parasitoids. The rapid development of resistance against these fungi limits the efficacy of chemical treatment. Damage caused by rice blasts must be prevented, especially in countries where agriculture is the primary industry [7]. Appressorium formation is a critical step in the infection cycle of *M. oryzae*. Therefore, inhibiting appressorium formation may help reduce damage from the disease. The presence of melanin is necessary for *M. oryzae* spore germination [8,9]. The appressorium’s color gradually deepens and the cell wall thickens as a result of melanin deposition and turgor pressure, thereby becoming a mature appressorium [10]. Biological control is observed as an environmentally friendly alternative for inhibiting *M. oryzae*, due to the many hazards caused by chemicals.

Plant-derived compounds can be effectively applied for disease prevention. Plant extracts can inhibit many of the plant diseases caused by fungi [11,12]; therefore, it would be appropriate to use plant-derived biomolecules for their antifungal properties. For example, *Azadirachta indica* extracts can effectively control *M. oryzae* [13]. *Azadirachta indica* leaf extracts and seeds effectively reduce the pathogen’s radial growth in vitro, as well as in controlling *M. oryzae* development and spread in rice plants. Citral is a naturally occurring compound that can disrupt *M. oryzae*, causing physiological changes and cytotoxicity by reducing sugar, soluble protein, chitinase activity, pyruvate, malondialdehyde, and cell-wall glucans. Comparative proteomic analysis has demonstrated that citral inhibits the proteins involved in oxidative phosphorylation, the TCA cycle (the tricarboxylic acid cycle), glycolysis, and translationally related pathways [11,12,13,14]. Compared with synthetic fungicides, natural chemicals isolated from plants are cheaper, readily available, and more cost-effective for use in developing countries [13,15,16,17].

*Aloe vera* is the world’s most widely recognized herbal medicine because it provides preservative, anti-inflammatory, and antifungal benefits [18]. *A. vera* extract has a strong inhibitory effect on *M. oryzae*. An aqueous *Aloe vera* leaf extract also has high antimicrobial activity against the mycelial growth of *M. oryzae*, which causes leaf blast [19]. *Aloe vera* extracts (25% *v*/*v*) exerted a 79.45% inhibitory effect against *Pyricularia grisea* from rice blast disease, but these extracts had no phytotoxic effects on seed germination, shoot height, root length, dry weight, seedling growth, or the seedling vigor index [4]. These studies are the first to investigate the antimicrobial effects of specific components of *Aloe vera* against *M. oryzae*. Aloe genus plants are rich in active ingredients and display a strong antifungal action against many fungi [20,21,22,23]. In a previous study, 1 mg/mL of aloesin was the MIC obtained against *Aspergillus niger, Candida albicans*, and *Penicillium funiculosum* [24]. Specifically, a test of 1000 mg/kg body weight/day of aloesin, a functional ingredient derived from *Aloe vera* extracts, demonstrated no adverse effects [25]. Inhibition of the formation of thiobarbituric acid reactive substances (TBARS) was demonstrated by aloesin-rich extracts, antimicrobial effects were found against bacterial and fungal strains, and antioxidant activity was also demonstrated, but no toxicity was observed for PLP2 cells [24]. The present study reports on the antifungal properties of aloesin, derived from *A. vera*, against *M. oryzae*. This study aims to evaluate the efficacy of aloesin in inhibiting conidia morphology and appressorium formation in *M. oryzae*. Based on proteomic analysis, we have elucidated the mechanism by which aloesin significantly inhibits *M. oryzae* spore germination and appressorium formation.

## 2. Materials and Methods

### 2.1. Magnaporthe oryzae

Guy 11 was obtained from the Agricultural Products Quality Safety Laboratory of the College of Agriculture at Guizhou University, Guiyang, China. Aloesin (95%) was obtained from the Xi’an Yunyue Biotechnology Co., Ltd., Xi’an City, Shanxi Province, China. The US3 multifunction digital microscope used in the test was manufactured by Shenzhen Love Science Digital Technology Co., Ltd., Shenzhen City, Guangdong Province, China. *M. oryzae* was cultured on potato sucrose agar (PSA) for two to three days at 28 °C. Purified *M. oryzae* was cultured on OTA medium and cultured at 25–28 °C for about 10 days. The mycelia were scraped off with sterilized slides; we then opened the lid of a Petri dish and the colony was introduced, followed by culturing in the light at about 25 °C, until a layer of grayish black conidium was produced on the surface of the colony. The strains with strong sporulation ability were then illuminated in the dark for 12 h using a fluorescent lamp on an ultra-clean workbench. The strain with weak sporulation ability was illuminated by black light until it produced a short villous and gray-black spore layer.

### 2.2. Effects of Aloesin on the Morphology of the Appressorium

*M. oryzae* spore morphology and germination were observed in water using a Hitachi SU8010 scanning electron microscope. *M. oryzae* spore morphology and appressorium formation in an aloesin solution were observed and recorded. Conidia were prepared by softly scratching the surface of 10-day-old fungal culture plates with a spatula. Low aloesin concentrations (625.00 μg/mL) were employed to research *M. oryzae* spore morphology and germination. A method of direct observation of the medium was used in this experiment. We cut a 0.5 cm × 0.5 cm medium containing the *M. oryzae* hypha layer and fixed it with 2.5% glutaraldehyde for 2 h. Then, *M. oryzae* hypha and spores were scanned with a Hitachi SU8010 scanning electron microscope. We then performed the following experimental operations: for cleaning with pH 7.2 phosphate buffer: we used ethanol in a series of 30%–50%–70%–85%–95%–100% (2 times) for step-by-step dehydration, each stage lasting 10–15 min (large blocks should be properly shaken to ensure clean dehydration). Intermediate fluid substitution was performed in two steps: the first step was soaking with a mixture of acetic (isoamyl) acetate and ethanol at 1:1 for 10 min, and the second step was soaking with isoamyl acetate for 10 min, then the mixture was appropriately shaken. For critical point drying, the sample was transferred to the sample basket, put into the dryer sample room at a pre-cooled critical point, the chamber was covered, and liquid carbon dioxide was injected until the sample was submerged. It was then heated up to 15 °C for 10 min, and the temperature was increased to 35 °C for gasification until the liquid had fully vaporized and slowly released the gas. It was only when the gas was exhausted that we opened the lid and took out the sample. Paste samples: General samples were pasted with double-sided tape; if the sample block was large and the electrical conductivity was poor, they were pasted with conductive adhesive. Observation by microscope was conducted after ion sputtering.

### 2.3. Inhibitory Effect of Aloesin on Hyphae and Appressoria of M. oryzae

A series of solutions containing different aloesin concentrations (5, 10, 20, 40, 80, 160, and 320 × 10^3^ μg/mL) were prepared. It was established that 95% aloesin was easily dissolved at a concentration of 320 × 10^3^ μg/mL. After melting PSA (at 40 °C) in an electric furnace, 1 mL aliquots of test solution were mixed with 9 mL PSA agar and added to 90 mm culture dishes. The test compound concentrations were 0.50, 1.00, 2.00, 4.00, 8.00, 16.00, and 32.00 × 10^3^ μg/mL. The untreated medium was used as the control. A piece of fungus disc was placed on each plate with six treatments (0.50, 1.00, 2.00, 4.00, 8.00, 16.00, and 32.00 × 10^3^ μg/mL) and the control. Three fungus discs were placed on each plate, and each treatment was performed in quadruplicate. *M. oryzae* was incubated at 28 °C for seven days, and the colony diameters of the tested fungus were recorded. The percentage inhibition of fungal growth was estimated, using the formula given by Ogbebor and Adekunle [26]:$$I=\frac{Dc-Dt}{Dc-Dd}\times 100\%$$
where *I* is the mycelial inhibition (%); *Dc* is the mycelial growth diameter in control (mm); *Dt* is the mycelial growth diameter in the treatment (mm), and *Dd* is the initial mycelial disc diameter (mm). According to the rate of inhibition (RI) of mycelial growth, aloesin’s EC_50_ (half maximal effective concentration) value was calculated using the IBM SPSS Statistics 19.0 software.

### 2.4. Enzyme Activity Analysis

The polyketide synthase (PKS), microcolumn ketone dehydratase (SCD), 1,3,6,8-tetrahydroxynaphthalene reductase (T4HR), 1,3,8-trihydroxynaphthalene reductase (T3HR), chain-shortening catalytic enzyme (Aayg1), and laccase (LAC) activities were determined according to the manufacturer’s instructions (Shanghai Jingkang Biological Engineering Co., Ltd., Shanghai City, China) for the samples, using the double antibody sandwich assay. The *M. oryzae* mycelium was treated with aloesin at different concentrations (250.00, 500.00, 1000.00, 2000.00, 4000.00, 8000.00, and 160,000.00 μg/mL), and non-treated PSA was used as a control. The *M. oryzae* mycelium weight was more than 50 mg. The sample was homogenized with phosphate buffer saline (0.01 mol/L, pH 7.2 to 7.4; sample:homogenate, 1:9) using a tissue homogenizer or it was ground manually. The homogenate was centrifuged at 1789× *g* for 15–30 min. The absorbance was measured using a microplate reader at 450 nm. The PKS, T4HR, SCD, T3HR, Aayg1, and LAC activities were calculated for the samples using a standard curve for each enzyme. All the tests were performed in triplicate.

### 2.5. Control Efficacy of Aloesin

The aloesin’s antifungal activity was investigated in vivo using the Becker–Ritt and Carlini method [27]. The rice variety used was “Dalixiang”. The wounding inoculation method uses mycelium blocks as inocula at a plant age of four to six leaves. After washing with water three times, the rice leaves were wounded using an anatomical needle to draw three wounds on the main vein. After spraying with 0.02% (*v*/*v*) Tween 20 solution, 9145.00 and 175.62 μg/mL of aloesin solutions were added separately to treat the wound, and mycelia (0.5 cm × 0.5 cm) were placed onto the injured part. After culturing in the dark (25 to 28 °C; 100% relative humidity (RH)) for 30 to 32 h, the leaves were placed in an illuminated incubator (25 °C, 100% RH) for 72 to 96 h. The incidence was measured using plaque size and color, and the aloesin’s antifungal activity was evaluated in vivo. The Tween 20 solution (0.02% *v*/*v*) was used as a positive control. Each group comprised two rice leaves, and all tests were performed in triplicate.

### 2.6. Data Analysis

An ANOVA conducted with Origin 2019b software was used to determine the least significant difference (*p* < 0.05) from the mean by analyzing the mean and standard deviation (SD). For models with significant fixed effects, differences between treatments were determined using the Origin 2021 at the α = 0.05 significance level, with an adjustment for Tukey’s HSD to control for family-wise error. All the measurements were repeated three times using duplicate samples.

### 2.7. Labeling Free Proteomic Analysis

#### 2.7.1. Protein Extraction

The pathogenic fungus *M. oryzae* was cultured on PSA medium (CK, Control; EC_50_, 175.62 μg/mL of aloesin; EC_95_, 625.00 μg/mL of aloesin) for five days. The sample (200 mg) was taken and 1 mL of PBS was added. The mixture was shaken at 56 °C for 10 min and centrifuged at 14,100× *g* for 5 min. The supernatant was discarded, and the sample was washed repeatedly three times. Each sample was ground individually in liquid nitrogen and lysed with 500 μL lysis buffer (8 M urea, 100 mM Tris-HCl, and 10% Protease Inhibitor Cocktail), followed by 2 min of ultrasonication on ice. The lysate was centrifuged at 12,000× *g* for 20 min at 4 °C, then the supernatant was transferred to a clean test tube. The protein concentration was determined using the Bradford method; the rest of the sample was frozen at −80 °C.

#### 2.7.2. Protein Digestion and Desalination

Reduced aliquots of proteins (100 g) were obtained from each sample after extraction. After incubating at 37 °C for 60 min, the solution was supplemented with 200 mM solution of dithiothreitol (DTT). To dilute the sample, 25 mM of ammonium bicarbonate (ABC) buffer was added four times. Next, trypsin (trypsin:protein =1:50) was introduced and the sample was left to incubate at a temperature of 37 °C for the entire night. On the following day, an additional 50 μL of 0.1% formic acid (FA) was introduced to halt the digestion process. To cleanse the C18 column, 100 μL of pure ACN was utilized and then subjected to centrifugation at a speed of 1200 rpm for a duration of 3 min. The column was rinsed once using 100 microliters of 0.1% formic acid and then subjected to centrifugation at a speed of 1200 revolutions per minute for a duration of 3 min. Following the replacement of the EP tube, the sample was introduced and subjected to centrifugation at a speed of 1200 rpm for a duration of 3 min. The column was rinsed two times with 100 μL of 0.1% formic acid and then spun at a speed of 1200 revolutions per minute for a duration of 3 min. The column was rinsed once using 100 microliters of water at a pH of 10. The EP tube was replaced and washed with 70% ACN. The eluents from each sample were mixed, freeze-dried, and kept at −80 °C until they were loaded.

#### 2.7.3. Peptide Identification Using Nano UPLC-MS/MS

The acquired peptide fractions were suspended with 20 μL of buffer A (0.1% FA, 2% ACN) and centrifuged at 12,000 rpm for 10 min. Briefly, 10 μL of the supernatants were injected into a nano UPLC-MS/MS system, consisting of a Nanoflow HPLC system (EASY-nLC 1000 system from Thermo Scientific, Waltham, MA, USA) and Orbitrap Exploris™ 480 mass spectrometer (ThermoFisher Scientific). The sample was loaded onto the Acclaim PepMap100 C18 column and separated using an EASY-Spray C18 column. The mass spectrometer was operated in positive ion mode (source voltage 2.1 KV), and the full MS scans were performed in the Orbitrap over the range of 300–1500 *m*/*z* at a resolution of 120,000. Following one full MS scan, the 20 most abundant ions with multiple charge states were selected for higher-energy collisional dissociation fragmentation for MS/MS scans. The *M. oryzae* database was used in this experiment. Proteome Discoverer 2.4 was used to process the MS/MS data.

#### 2.7.4. Proteomic Data Analysis

Proteome Discoverer (version 2.4, Thermo Fisher Scientific) was used to analyze all RAW files. The following parameters were used for identification: precursor ion mass tolerance, ±15 ppm; fragment ion mass tolerance, ±20 mmu; max missed cleavages, 2; static modification, carboxy amidomethylation (57.021 Da) of Cys residues; and dynamic modifications, oxidation modification (+15.995 Da) of Met residues. Peptide spectral matches were validated, and fault detection rate (FDR)verification was performed to remove peptides and proteins with an FDR of >1%. Median normalization was performed on the original data to eliminate any errors caused by the experiment. Data with more than 50% of null values were filtered in the sample. The missing values were imputed into the filtered data using the Perseus algorithm. Differentially expressed proteins (DEPs) satisfied the following conditions: average ratio-fold change > 1.2 and *p*-value < 0.05 (Student’s *t*-test). Gene ontology (GO) annotation of the proteome was derived from the GO database http://www.ebi.ac.uk/QuickGO/ (accessed on 23 September 2022) [28]. The pathway analysis was performed using the Kyoto Encyclopedia of Genes and Genomes (KEGG) pathway (http://www.kegg.jp/kegg/pathway.html) (accessed on 23 September 2022) [28] to test the enrichment of the different proteins against all identified proteins by hypergeometric distribution.

## 3. Results

### 3.1. Effects of Aloesin on the Morphology of Hyphae and Appressorium of M. oryzae

The antimicrobial activity of aloesin on *M. oryzae* spores revealed that 625.00 μg/mL aloesin caused one or both ends of the *M. oryzae* spores to break or rupture at 12 h, and the hyphae were weaker (Figure 1(B1)) than those of the control (Figure 1(A1)). When treated with 625.00 μg/mL aloesin, the *M. oryzae* spores collapsed compared to the polar growth of *M. oryzae* spores in the control, reducing the level of polar growth on spores and the formation of appressoria (Figure 1(B2)) compared to the control (Figure 1(A2)).

### 3.2. Inhibitory Effect of Aloesin on Hyphae and Appressorium of M. oryzae

The different aloesin concentrations clearly affected spore morphology, specifically appressorium formation. Only a few spores were germinated, mycelial growth was weak, and an appressorium was not formed with an aloesin concentration. We established that 625.00 μg/mL of aloesin caused the rupturing of 92.30 ± 4.26% of spores and a rate of appressorium formation of 4.74 ± 1.00% at 12 h. The percentage of appressorium formation decreased, and the spore rupturing increased, as the aloesin concentration increased above 625.00 μg/mL of aloesin (Figure 2A). The EC_50_ and EC_95_ values of the appressorium formation rate were 175.62 μg/mL and 625.00 μg/mL, respectively. According to the results, aloesin inhibited appressorium formation significantly better than it did mycelial growth in *M. oryzae*. The inhibition rate also increased as the aloesin test concentration increased. Specifically, 32.00 × 10^3^ μg/mL of aloesin revealed the lowest inhibition rate of 15.64% on day two and the highest inhibition rate of 74.45% on day seven, representing an increase in the inhibition rate of 58.81%. The inhibitory effect was continuously enhanced at the same concentration; the EC_50_ value was 9145.00 μg/mL after four days (Figure 2B).

Although 175.62 μg/mL of aloesin caused the *M. oryzae* colony to fade, mycelium growth had no apparent inhibitory effect, and the dense mycelium layer became thinner (Figure 3).

### 3.3. Effect of Aloesin on Enzyme Activity of M. oryzae

Regarding the inhibitory effect of aloesin on enzyme activity, PKS indicated the highest activity of 439.09 U/L in the control and the lowest activity of 296.36 U/L at 16.00 × 10^3^ μg/mL of aloesin, representing an increase in inhibition of 142.73 U/L and an inhibition rate of 32.51%. Laccase activity decreased by 52.70 U/L, with a 43.38% inhibition rate. The inhibition rate was 33.04%, with Aayg1 activity decreasing by 27.67 U/L and SCD activity decreasing by 19.88 U/L. These results indicated that aloesin inhibits the melanin synthase Aayg1, PKS, and LAC of *M. oryzae*, which is possibly one of the causes of colony fading (Figure 4).

### 3.4. Control Efficacy of Aloesin

The virulence regression equation of aloesin to the inhibition rate of the hyphae of *M. oryzae* is Y = 40.855 + 0.001X (R-value = 0.9434), where EC_50_ = 9145.00 μg/mL. The virulence regression equation of aloesin to the inhibition rate of the appressorium of *M. oryzae* is Y = 1.911X + 0.7104 (R-Value = 0.8801), EC_50_ = 175.26 μg/mL. The inoculation results showed that the vaccination symptom in the CK was the obvious brown oval lesions and dark areas in the centers of disease spots. The results of inoculation revealed similar disease symptoms at 9145.00 μg/mL and 175.62 μg/mL of aloesin (Figure 5), the inoculated sites of the leaves showed only slight discoloration, and the disease spots were not obvious, indicating that 175.62 μg/mL of aloesin had an obvious anti-invasion effect on *M. oryzae*. It was also indirectly proved that the invasion of *M. oryzae* was closely related to the production of appressoria.

### 3.5. Labeling Free Proteomic Analysis

The information was converted using centered-log and regularized-log proportions. The heat map of the sample-to-sample distance produced in the principal component analysis of the top 300 variable proteins showed high levels of similarity among all three replicates (Figure 6A). According to the principal component analysis, there were no notable variances in gene expression between the biological duplicates. The proteomic analysis data were dependable and satisfied the requirements for differential expression analysis (Figure 6B).

The proteomic analysis identified 51 differentially expressed proteins (DEPs), with 32 proteins upregulated and 19 downregulated at 175.62 μg/mL of aloesin (Figure 7A). We also identified 227 DEPs, with 127 proteins upregulated and 100 proteins downregulated at 625.00 μg/mL of aloesin (Figure 7B). This proteomic analysis displayed significantly different cell expression levels compared to the non-treated cases. In our study, 175.62 μg/mL of aloesin reduced the actin (MGG_03982) expression levels (Figure 7C).

GO analysis of proteins from each cluster revealed that the downregulated proteins exhibited metabolic and biosynthetic functions at 625.00 μg/mL of aloesin, including the carbon metabolic process, canonical glycolysis, malate metabolic process, organic substance metabolic process, microtubule-based process, pyruvate biosynthetic process, and polyamine biosynthetic process between aloesin-treated and control samples, presumably reducing the amount of energy and materials that will be provided to support the basic requirements for appressorium formation by *M. oryzae*. However, the upregulated proteins are involved in mitochondrion organization, cellular manganese ion homeostasis, malonyl-CoA biosynthesis, cytoplasmic translational elongation, intracellular signal transduction, and fungal-type cell wall biogenesis, suggesting that these molecular events underlie the change in spore morphology (Figure 8A). The GO analysis of proteins from each cluster revealed that the downregulated proteins exhibited metabolic and biosynthetic functions at 175.62 μg/mL of aloesin, including protein translation and the cellular amino acid metabolic process, presumably reducing the biological activity and materials needed to support the basic requirements for protein synthesis in *M. oryzae*. However, the upregulated proteins form part of the polysaccharide catabolic and carbohydrate metabolic processes, suggesting that these molecular events degrade nutrients (Figure 8B).

The enriched GO analysis results of EC_50_ revealed that the most upregulated gene categories were of ribosomes, while the most downregulated gene categories were of cysteine and methionine metabolism. The enriched gene ontology analysis results of EC_95_ indicated that the most upregulated gene categories were glycosphingolipid biosynthesis, the globo and isoglobo series, and starch and sucrose metabolism, while the most down-regulated gene categories were ribosome and fatty acid degradation (Figure 8C).

## 4. Discussion

Rice blast disease, which is caused by the pathogenic fungus *M. oryzae*, causes widespread rice damage and yield loss [29]. Biological control technologies that reduce the use of chemical fungicides have recently received increased research attention. The prevention and control of rice blast disease is considered paramount for safeguarding the security of rice production. It is important to understand the biological characteristics of conidia germination, appressorium formation, and the disease development of *M. oryzae* for controlling rice blast [30,31]. Appressorium formation is a critical step in the infection cycle of *M. oryzae*. Therefore, inhibiting appressorium formation may help reduce damage from the disease. The presence of melanin is necessary for *M. oryzae* spore germination. In this study, aloesin effectively inhibited melanin synthesis and appressorium formation. Aloesin has a good inhibitory effect on the melanin synthase Aayg1, PKS, and LAC of *M. oryzae.* Aloesin exhibited high inhibitory effects on appressorium formation (the EC_50_ value was 175.26 μg/mL) and low inhibitory effects on the mycelial growth of *M. oryzae* (the EC_50_ value was 9145.00 μg/mL). The inhibitory effect was enhanced with increasing exposure from two to seven days. These results demonstrated that aloesin could effectively control the growth and proliferation of. *M. oryzae*. Additionally, different concentrations of aloesin affected spore morphology and appressorium formation, as was evident from the microscopic examination. At aloesin concentrations exceeding 312.50 μg/mL, no appressoria were produced, only a small number of spores germinated, mycelial growth was weak, and the average diameter of the hypanthium was less than that of the control. Appressorium formation is required for the pathogenicity of *M. oryzae*. Compared to spore germination in the control group, the spores were damaged at one or both ends in response to treatment with aloesin, reducing the number of germinating spores and appressorium formation. The formation of appressoria and their normal functioning are critical for *M. oryzae* adhesion and the penetration of host cells [32,33,34]. DHN-melanin plays an important role in the pathogenicity of appressorium, and its synthesis pathway is as follows: malonyl-CoA is found in polyketide synthase (PKS) and catalyzes the synthesis of 1,3,6,8-tetrahydroxynaphthalene (T4HN), then, with the action of a specific reductase, it is restored to 2,4-dihydro-3,6,8-trihydroxy-1-naphthalone (scytalone, SCY). SCY is dehydrated to form 1,3,8-trihydoxynaphthalene (1,3,8-THN), followed by 2,4-dihydro-3,8-dihydroxy-1-naphthalene (vermelone, VER) under the action of a second reductas and is then dehydrated to form 1,8-dihydroxynaphthalene (1,8-DHN). Finally, 1, 8-DHN polymerizes to produce DHN-melanin under the action of laccase [35]. The enzyme activity analysis results indicated that aloesin inhibited polyketolase (PKS), laccase (LAC), and chain-shortening catalytic enzyme (Aayg1) activities; these are key enzymes in melanin synthesis. The inhibition rate by aloesin of PKS, LAC, and Aayg1 activity was 32.51%, 33.04%, and 43.38%, respectively. The inhibitory effect of aloesin on enzyme activity was strongest in Aayg1, LAC, and PKS, and was least effective regarding T4HR activity. Aayg1, LAC, and PKS are important enzymes for melanin synthesis in *M. oryzae* spores and the appressoria [35]. Thus, aloesin exhibited high inhibitory effects on appressorium formation.

The actin cytoskeleton is believed to play a key role in the development and pathogenesis of *M. oryzae* [36,37,38,39]. All eukaryotes contain actin, which is essential in many cellular processes [31,40,41,42]. The actin cytoskeleton assembles as patches, cables, and rings in filamentous fungi [43,44,45,46]. The development and pathogenesis of *M. oryzae* largely depend on the actin cytoskeleton [37]. Xu discovered that onion epidermal cells accumulate a large amount of actin in the appressorium at the point of contact between rice blast fungus and the onion surface. An appressorium penetration peg is formed when *M. oryzae* septins polymerize into a heteromeric ring to scaffold a toroidal f-actin network, breaching the leaf surface. Reduced actin bundle formation inhibits plant cell polar growth, vesicle transport, and hyphal expansion [47]. This experiment revealed that aloesin could induce the downregulation of actin at a concentration of 175.26 μg/mL. Appressorium formation, rather than that of mycelia, determines *M. oryzae* invasion, and PKS and actin are important for melanin synthesis in the appressorium [30,35,36]. In particular, aloesin could induce the downregulation of actin at concentrations of 175.26 μg/mL. Reducing actin bundle formation would prevent polar growth, vesicle transport, and hyphal expansion in *M. oryzae*.

## 5. Conclusions

This study is the first to report that aloesin inhibits the appressorium germination and enzyme activity of melanin synthase Aayg1, PKS, and LAC of *M. oryzae*; the mechanism of aloesin could induce the downregulation of actin. The results presented herein support the proposition that aloesin controls mostly spore germination and appressorium formation, thereby reducing blast spot production. Aloesin is safe and environmentally friendly compared to synthetic fungicides. However, additional studies are needed to identify and characterize the active antifungal compounds in the extracts and determine their roles in rice blast disease control. Molecular docking and experimental verification will be performed for the mechanism investigation.

## Figures and Tables

**Figure 1 microorganisms-11-02395-f001:**
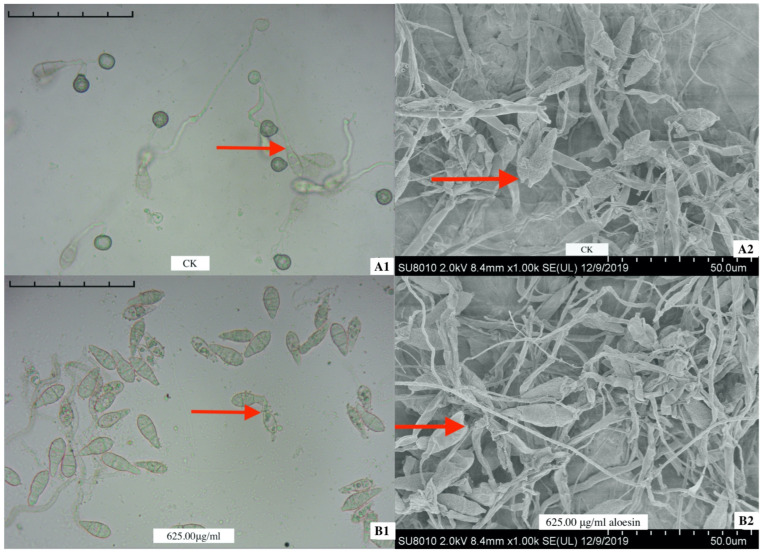
The spores of *M. oryzae* collapsed (indicated by an arrow) when aloesin was used against the *M. oryzae* spores (12 h). (**A1**,**A2**) Control. (**B1**,**B2**) 625.00 μg/mL aloesin. Bar = 50.00 µm.

**Figure 2 microorganisms-11-02395-f002:**
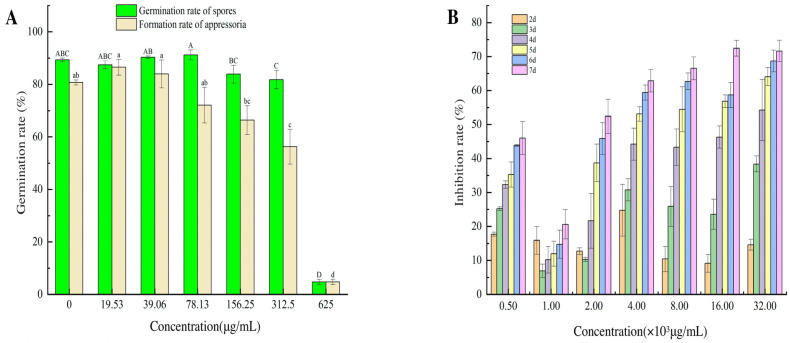
The inhibitory effect of aloesin on the hyphae and appressoria of *M. oryzae*. (**A**) Inhibition rate of aloesin against the morphology and germination of *M. oryzae* spores under microscopic examination (12 h). (**B**) Inhibition rate of aloesin against mycelium growth in *M. oryzae* (2−7 d).

**Figure 3 microorganisms-11-02395-f003:**
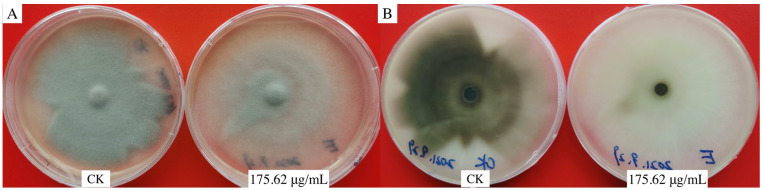
Inhibitory effect of 175.62 μg/mL of aloesin against *M. oryzae* Guy11. (**A**) Colony front view; (**B**) colony back view.

**Figure 4 microorganisms-11-02395-f004:**
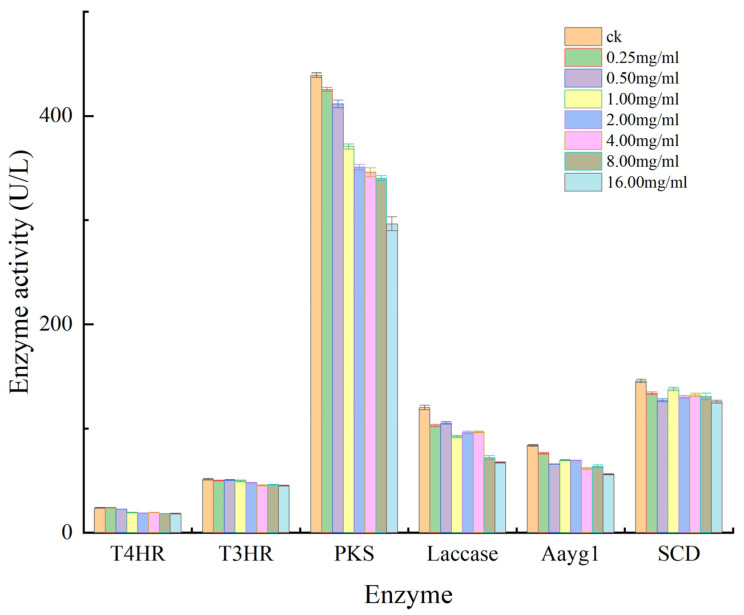
The enzyme activity effect of aloesin against *M. oryzae*.

**Figure 5 microorganisms-11-02395-f005:**
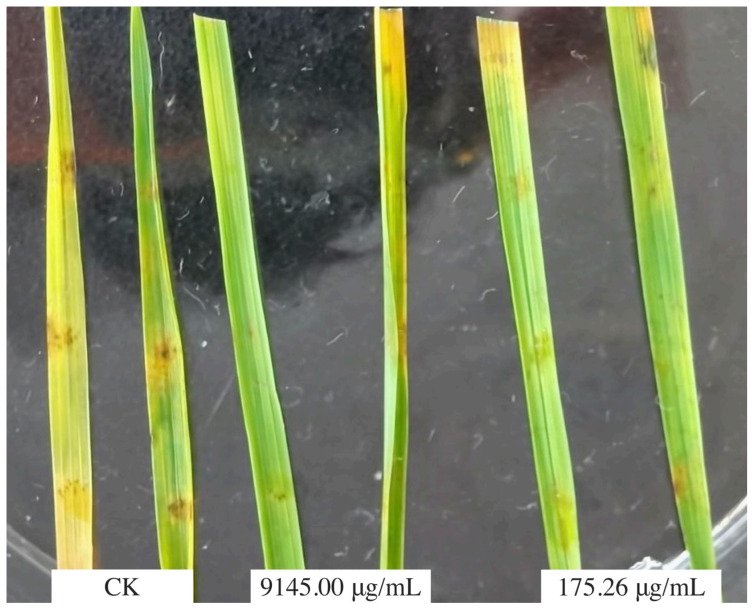
Inoculation effect of aloesin against *M. oryzae* by the inoculation method.

**Figure 6 microorganisms-11-02395-f006:**
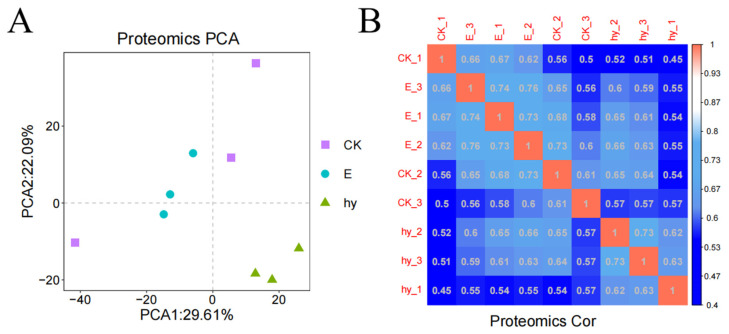
Quality control chart of the protein. Data were transformed with the centered-log and regularized-log ratios (**A**). The sample-to-sample distance heat map exhibited good similarity between all three replicates (**B**). CK, Control; EC_50_, 175.62 μg/mL of aloesin; EC_95_, 625.00 μg/mL of aloesin.

**Figure 7 microorganisms-11-02395-f007:**
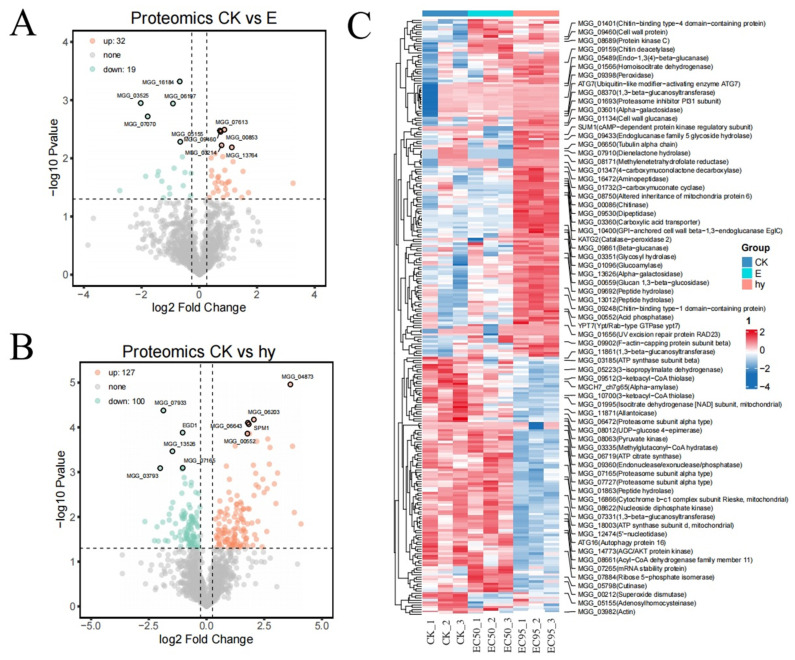
Volcano plot and heatmaps of DEPs in *M. oryzae* between the aloesin-treated and control samples. (**A**,**B**) Volcano plot illustrating the upregulated (red dots) and downregulated (blue dots) DEPs and the normally expressed proteins (gray dots). (**C**) Heatmap of up-regulated DEPs and down-regulated DEPs. CK, Control; EC_50_, 175.62 μg/mL of aloesin; EC_95_, 625.00 μg/mL of aloesin.

**Figure 8 microorganisms-11-02395-f008:**
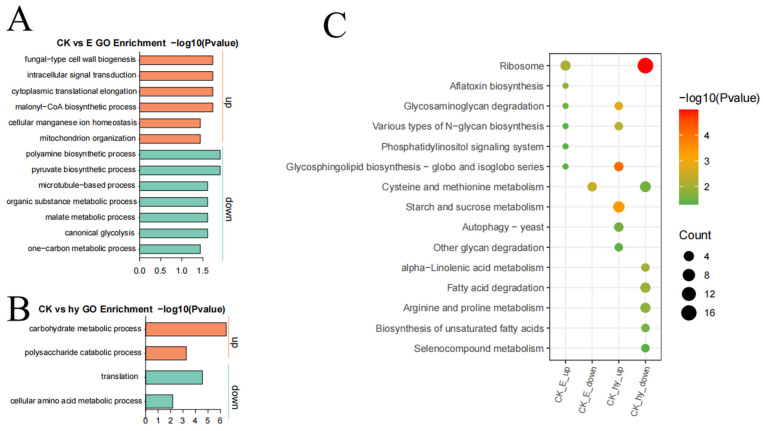
KEGG enrichment analysis of the proteins in *M. oryzae* between the aloesin-treated and control samples. (**A**,**B**) Pathway analysis showing the upregulated (red dots) and downregulated (green dots) DEPI. (**C**) Differentially expressed proteins induced by aloesin treatment, shown as an enriched gene ontology analysis of up-regulated DEPs and an enriched gene ontology analysis of down-regulated DEPs. padj: adjusted *p*-value. CK, Control; EC_50_, 175.62 μg/mL of aloesin; EC_95_, 625.00 μg/mL of aloesin.

## Data Availability

Data will be made available upon request.

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
