# Peer review of "Natural Product Aloesin Significantly Inhibits Spore Germination and Appressorium Formation in Magnaporthe oryzae"

_microorganisms, 2023, doi:10.3390/microorganisms11102395_

Round 1
Reviewer 1 Report
As after careful revision, the entire manuscript described well with strong scientific methodologies and supportive references. The abstract, Introduction, Methods, results and discussion were very clear to the research objectives.
But in conclusion there was a repetition of the following sentence to be removed.
"However, additional studies are needed to identify and characterize the active antifungal compounds in the extracts and determine their roles in rice blast disease control. However, additional studies are needed to 381 identify and characterize the active antifungal compounds in the extracts and determine their roles in rice blast disease control"
Need mild language correction is needed to avoid the "environmentally friendly" in the end of first para of introduction part. Also I feel title also need correction as Natural product aloesin instead Naturally products aloesin...
Biological control is observed as an environmentally friendly alternative for inhibiting M. 42 oryzae due to the many hazards of chemicals
Is the protein sequence deposited in GenBank?
Need a mild language corrections for publication betterment
Reviewer 2 Report
The article presents an important contribution regarding the use of natural products to combat fungal diseases in rice instead of the use of pesticides. I consider that the article is ready for publication.
Reviewer 3 Report
Dear Authors
The current manuscript entitled “Naturally products aloesin significantly inhibits spore germi nation and appressorium formation in Magnaporthe oryzae” discusses the effects of aloesin against Magnaporthe oryzae and suggested that aloesin had a high inhibitory effect on appressorium for mation in the spores. There are certain opportunities for further improvement, please find my suggestions below.
1. The manuscript need to be checked by a native speaker. For example the beginning of introduction “Magnaporthe oryzae is a destructive pathogen that is a major concern in rice production [1,2]. Rice blast, caused by M. oryzae, is considered a principal disease impacting rice [3,4]” can be written as “Magnaporthe oryzae remains the major pathogen in rice crop, affecting the production negatively across the different geographical regions”.
2. Line 31-35, Please include a reference for the claim.
3. Introduction need to be focused on Magnaporthe oryzae and basic mechanism of spore germination towards appressorium development. Further some information can be included about Aloesin and central idea of present work.
4. Obtaining Magnaporthe oryzae (Guy 11) spores in enough quantity is challenging, please include more details how spores were obtained? Did you receive the spores from “Agricultural Products Quality 77 Safety Laboratory of the College of Agriculture at Guizhou University, Guiyang” or fungal cultures were obtained? Which media was used to cultivate and the conditions, for example in dark or light, temperature and humidity?
5. Line 79- Aloesin (95%) was obtained from Xi’an Yunyue Biotechnology Co., Ltd, Kunming City, Yunnan Province, China. What is the solvent and how the extracts were obtained? Please include more details.
6. Line 90-92, The method of direct observation on medium was used in this experiment. Cut a 0.5cm*0.5cm piece of blast fungus together with the medium and fix it with 2.5% glutaraldehyde for 2 h. Then perform the following experimental operations: Clean with pH 7.2 phosphate buffer. Please rewrite these sentences to make it clearer.
7. Line 98-102, “Critical point drying: the replaced sample is transferred into the sample basket, put into the pre-cooled critical point dryer sample room, cover the chamber, inject liquid carbon dioxide, as the sample is submerged, heat up to 15℃ for 10 minutes, and then heat up to 35℃ to vaporize it, observe the liquid fully vaporized and slowly release the gas, and only when the gas is exhausted can the cover be opened for sampling”. Please rewrite these sentences as well.
8. There is no data for spores counting before inoculation?
9. Figure 1, the germination was checked for 12hr only? Sometimes the presence of biocompounds delays the spores germination. I would suggest checking for 24hr as well.
10. Figure 1, “The spores of M. oryzae collapsed (arrow) of aloesin against the M. oryzae spores”. The figure shows only couple of spores collapsed while other spores look healthy, and some of them have germination tube as well. Only difference I could see clearly is, control sample have appressorium while treatment does not have any.
11. Figure 5 is most important for the current study, please explain it in details in methods and result section. For example, the rice plant variety, age of seedling, how leaves were maintained, number of spores inoculated per wound and disease development. Additionally, it should be confirmed if the symptoms are Blast diseases. I am not sure if figure 5 has disease development, the wounds looks like normal necrosis instead of rice blast disease.
12. Conclusion should be very concise, passing the key message of the present work.
Thank you
Dear Authors
The manuscript need a thorough revision by a native speaker. There are certain sentences through out the manuscript which need to reframe.
Thank you
Reviewer 4 Report
The manuscript "microorganisms-2583915" presents interesting data on the effect of the plant-derived substance aloesin on the phytopathogenic fungus Magnaporthe oryzae. I think that this work can be accepted for publication in Microorganisms after a major revision.
Basic remarks:
1. The authors describe the effect of aloesin on the fungus development at different concentrations, including very high concentrations >5 mg/ml. The solubility of aloesin in aqueous media is not so high. At the same time, the authors do not describe which preparation of aloesin they used in their work. This leads to a misunderstanding of what exactly had an effect on the fungus.
2. The manuscript lacks a proper description and discussion of the results of the experiment regarding "Control Efficacy of Aloesin".
Minor remarks:
1. lines 13-18: three sentences repeat each other in meaning. Change the style of the text so that there are no repetitions.
2. line 16: “Laccase” in this place should be with a small letter.
3. The abstract contains the abbreviations “PKS”, “LAC”, and “Aayg1”. It is better to avoid this or give their decoding.
4. Unfortunate style of sentences "Rice blast, caused by M. oryzae, is considered a principal disease impacting rice" (line 29), "Aloesin reduced the actin (MGG_03982) expression levels at 175.62 μg/mL of aloesin" (lines 286-287) and others like that. Correct so that there are no repetitions of words.
5. lines 40-42: Provide a reference to a literary source for this sentence.
6. in the text of the manuscript, the abbreviation "A." used for both "Azadirachta" and "Aloe". Correct to make it clear in each case what "A" means.
7. Add a description and characterization of aloesin to the Introduction section.
8. line 86: Add a description of the aloesin preparation used in the work (manufacturer, purity and other characteristics).
9. lines: 123-125: Why were these particular enzymes chosen? According to the literature, aloesin inhibits tyrosinase. Why was the effect of aloesin on the tyrosinase of Magnaporthe oryzae not investigated?
10. lines 128: What solvent was used to prepare different concentrations of aloesin?
11. lines 129 and 142: Aloesin is poorly soluble in the aquatic media. How were solutions prepared with high concentrations of aloesin?
12. line 241: How did the authors calculate that the EC50=9145 mkg/mL? Figure 2B shows that at 16 mg/mL, aloesin inhibits less than 50%. In addition, Figure 2B shows that at aloesin concentrations greater than 4 mg/mL, mycelial growth stops after 2 days. Was it like that?
13. lines 264-266: Authors should provide for this part of the manuscript a more detailed description of the results and their discussion in the Discussion section.
14. line 274: What allows authors to discuss gene expression?
15. line 335: "melanin synthesis" - This manuscript does not provide data on the content of melanin.
16. line 336: How are PKS, LAC, and Aayg1 related to melanin synthesis? Add this information to the text.
17. lines 378-382: the same sentence is repeated twice "However, additional studies are needed to identify and characterize the active antifungal compounds in the extracts and deter mine their roles in rice blast disease control."
Round 2
Reviewer 3 Report
Dear Authors
Thank you for your patience while answering my queries and providing a revised version. The current version has been significantly improved.
Especially I appreciate the details of materials and methods improvement. Rice blast is very important disease which needs attention to develop the biological control to support the sustainable crop protection practices.
I declare that all my suggestions have been included in the manuscript and i do not have any further query regarding this manuscript.
Thank you
Regards
Reviewer 4 Report
1. Between references 23 and 24, state that aloesin is a metabolite of Aloe vera and the chemical structure is a C-glycosylated 5-methylchromone. And if it has been previously shown, it has antifungal properties. - Including references to works where this was previously described.
2. In section 2.3, after the first sentence, describe the preparation of aloesin solutions. Indicate what manipulations were used for this. If aloesin was easily dissolved at a concentration of 32 mg/ml, be sure to indicate this in the text, since this does not coincide with the reference information on the solubility of aloesin.
3. The authors write, “The heat map of the sample-to-sample distance in the principle component analysis of the top 300 variable proteins showed a high level of similarity among all three replicates (Figure 6A). According to the principal component analysis, there were no notable variances in gene expression between the biological duplicates.” Caption for Figure 3: “Figure 6. Quality control chart of protein. Data were transformed with the centered-log and regularized-log ratios (A). The sample-to-sample distance heat map exhibited good similarity between all three replicates (B). CK, Control; EC50, 175.62 μg/mL of aloesin; EC95, 625.00 μg/mL of aloesin." - I don't understand what allows the authors to discuss gene expression.
